# A divisive model of evidence accumulation explains uneven weighting of evidence over time

Waitsang Keung [1✉], Todd A. Hagen[1] & Robert C. Wilson[1,2]

Divisive normalization has long been used to account for computations in various neural processes and behaviours. The model proposes that inputs into a neural system are divisively normalized by the system's total activity. More recently, dynamical versions of divisive normalization have been shown to account for how neural activity evolves over time in value-based decision making. Despite its ubiquity, divisive normalization has not been studied in decisions that require evidence to be integrated over time. Such decisions are important when the information is not all available at once. A key feature of such decisions is how evidence is weighted over time, known as the integration kernel. Here, we provide a formal expression for the integration kernel in divisive normalization, and show that divisive normalization quantitatively accounts for 133 human participants' perceptual decision making behaviour, performing as well as the state-of-the-art Drift Diffusion Model, the predominant model for perceptual evidence accumulation.

[1] Department of Psychology, University of Arizona, Tucson, AZ 85719, USA. [2] Cognitive Science Program, University of Arizona, Tucson, AZ 85719, USA.
✉email: wkeung@email.arizona.edu

Divisive normalization has been proposed as a canonical computation in the brain[1]. In these models, the firing rate of an individual neuron is computed as a ratio between its response to an input and the summed activity of a pool of neurons receiving similar inputs. For example, the activity of a visual cortex neuron $f_i$ responding to an input $u_i$ can be computed as the input divided by a constant $S$ plus a normalization factor—the sum of inputs received by the total pool of neurons[1]:

$$f_i = \frac{u_i}{S + \sum_j u_j} \tag{1}$$

Divisive normalization models such as described in Eq. (1) have been used successfully to describe both neural firing and behavior across a wide range of tasks—from sensory processing in visual and olfactory systems[2–5], to context-dependent value encoding in premotor and parietal areas[6]. For example, in the visual domain, divisive normalization explains surround suppression in primary visual cortex, where the response of a neuron to a stimulus in the receptive field is suppressed when there are additional stimuli in the surrounding region[7]. Analogously, in economic decision making, divisive normalization explains how activity in parietal cortex encodes the value of a choice option relative to other available alternatives instead of the absolute value[6]. More recently, dynamic versions of divisive normalization models have been used to describe how neural activity in economic decision making tasks evolves over time[8,9].

Despite the success of divisive normalization models, they have never been studied in situations that require evidence to be integrated over time. Such evidence accumulation is important in many decisions when we do not have all the information available at once, such as when we integrate visual information from moment to moment as our eyes scan a scene. In the lab setting, evidence accumulation has typically been studied in perceptual decision making tasks over short periods of time. In one such task, called the Poisson Clicks Task[10], participants make a judgment about a train of auditory clicks. Each click comes into either the left or right ear, and at the end of the train of clicks participants must decide which ear received more clicks. The optimal strategy in this task is to count, i.e., integrate, the clicks on each side and choose the side with the most clicks.

A key feature of any evidence accumulation strategy is how evidence is weighted over time, which is also known as the kernel of integration. For example in the optimal model of counting, each click contributes equally to the decision, i.e., all clicks are weighed equally over time. In this case, the integration kernel is flat—the weight of every click is the same. While such flat integration kernels have been observed in rats and highly trained humans[10], there is considerable variability across species and individuals. For example, Yates and colleagues[11] showed that monkeys exhibit a strong primacy kernel, in which early evidence is overweighed. An opposite, recency kernel, where early evidence is under weighed, was observed in humans[12,13]. Recently, in a large scale study of over a hundred human participants, we find that different people use different kernels with examples among the population of flat, primacy, and recency effects. Intriguingly, however, the most popular kernel in our experiment is a bump shaped kernel in which evidence in the middle of the stimulus was weighed more than either the beginning or the end[14].

In this work, we show how dynamic divisive normalization[8] can act as a model for evidence accumulation in perceptual decision making. We provide theoretical results for how the model integrates evidence over time and show how dynamic divisive normalization can generate all of the four integration kernel shapes: primacy, recency, flat, and (most importantly) the bump kernel which is the main behavioral phenotype in our task[14]. In addition, we provide experimental evidence that divisive normalization can quantitatively account for human behavior in an auditory perceptual decision making task. Finally, with formal model comparison, we show that divisive normalization fits the data quantitatively as well as the state-of-the-art Drift Diffusion Model (DDM), the predominant model for perceptual evidence accumulation, with the same number of parameters.

## Results

**A divisive model of evidence accumulation.** Our model architecture was inspired by the dynamic version of divisive normalization developed by Louie and colleagues to model neural activity during value-based decision making[8]. We assume that the decision is made by comparing the activity in two pools of excitatory units, $R_{\text{left}}$ and $R_{\text{right}}$ (Fig. 1a). These pools receive time-varying input $C_{\text{left}}$ and $C_{\text{right}}$. In the Clicks Task (below), these inputs correspond to the left and right clicks, more generally they reflect the momentary evidence in favor of one choice over the other. An inhibitory gain control unit $G$, which is driven by the total activity in the excitatory network, divisively inhibits the $R$ unit activity. The time-varying dynamics of the model can be described by the following system of differential equations:

$$\tau_R \frac{dR_i}{dt} = -R_i + \frac{C_i}{1 + G} \tag{2}$$

$$\tau_G \frac{dG}{dt} = -G + \omega_I \sum_{i=1}^{N} R_i \tag{3}$$

A decision is formed by comparing the difference in activity $\delta$ between the two $R$ units

$$\delta = R_{\text{left}} - R_{\text{right}} \tag{4}$$

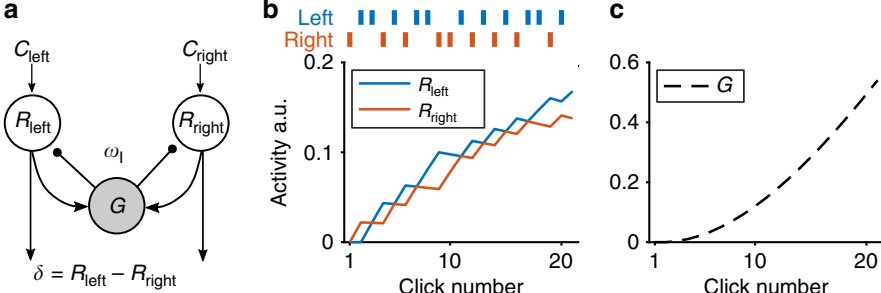

**Fig. 1 Dynamic divisive normalization schematic and simulated model dynamics. a** Schematic of dynamic divisive normalization model. The two excitatory $R$ units integrate punctate inputs $C$ respective to left and right. The inhibitory $G$ unit receives the sum of the two $R$ unit activity weighted by $\omega_I$, and in turn divisively normalizes the input to $R$. **b, c** Results of the model activity simulated with $\tau_R = 2.27$, $\tau_G = 11.10$, and $\omega_I = 36.20$.

Example simulated dynamics of the $R$ and $G$ units for punctate inputs (of the form used in the Clicks Task) are shown in Fig. 1b, c. The model has three free parameters: $\tau_R$, $\tau_G$, and $\omega_I$. As is clear from this plot, the $R$ unit activity integrates the input, $C$, over time, with each input increasing the corresponding $R$ unit activity. In addition, closer inspection of Fig. 1b reveals that the inputs have different effects on $R$ over time—for example, compare the effect of the first input on the right, which increases $R_{\text{right}}$ considerably, to that of the last input on the right, which increases $R_{\text{right}}$ much less. This suggests that the model with these parameter settings integrates evidence over time, but with an uneven weighting for each input.

**Divisive normalization generates different kernel shapes.** How can we quantify the integration kernel—how much each piece of evidence weighs—given by a circuit that generates divisively normalized coding? We integrate the set of differential equations to provide an explicit expression for the integration kernel. We first consider the evolution of the difference in activity, $\delta$, over time. In particular, from Eqs. (2) and (4), we can write

$$\tau_R \frac{d\delta}{dt} = -\delta + \frac{\Delta C}{1 + G} \tag{5}$$

where $\Delta C$ is the difference in input,

$$\Delta C = C_{\text{left}} - C_{\text{right}} \tag{6}$$

We can then integrate Eq. (5) to compute the following formal solution for $\delta$ as a function of time (for details of derivation, see Methods section):

$$\delta(t) = \int_0^t K(t, t') \Delta C(t') dt', \text{ where } K(t, t') = \frac{1}{\tau_R} \frac{\exp(-(t - t')/\tau_R)}{1 + G(t')} \tag{7}$$

This expression shows explicitly that the activity of the network acts to integrate the inputs $\Delta C$ over time, weighing each input by the integration kernel function $K(t, t')$. Importantly, $K(t, t')$ represents the degree to which evidence $\Delta C$ at time $t'$ contributes to the decision.

While clearly not a closed form expression for the integration kernel (notably $K(t, t')$ still depends on $G(t)$), Eq. (7) gives some intuition in how evidence is accumulated over time in this model. In particular, the kernel can be written as a product of two factors: an exponential function (Fig. 2a) and the inverse of the $G$ activity (Fig. 2b). The exponential function is increasing over time, and since $G$ is increasing with time (Fig. 1c), the inverse of $G$ is decreasing over time. Under the right conditions, the product of these increasing and decreasing functions can produce a bump shaped kernel, Fig. 2c.

More intuitively, we can consider the integration kernel as being affected by two processes: the leaky integration in $R$ and the increasing inhibition by $G$. If we consider the start of the train of clicks when $G$ is small, the model acts as a leaky integrator (Eq. (2)), which creates a recency bias since earlier evidence is "forgotten" through the leak. Over time, as $G$ unit activity increases, $G$ exerts an increasing inhibition on $R$, and when inhibition overcomes the leaky integration, later evidence was weighed less than the preceding evidence.

These intuitions suggest that the shape of the integration kernel is determined by a balance between how fast the leaky integration in $R$ happens (the rate of $R$) and how fast the inhibitory $G$ activity grows (the rate of $G$). These two rates are determined by the inverse of the time constants $\tau_R$ and $\tau_G$ respectively—i.e., when $\tau$ is large, the rate is slow. The balance between the rate of $R$ and the rate of $G$ can then be described as the ratio $\tau_R/\tau_G$—i.e., when $\tau_R$ is larger than $\tau_G$, $R$ activity is slower than $G$ activity, and similarly; when $\tau_R$ is smaller than $\tau_G$, $R$ activity is faster than $G$ activity.

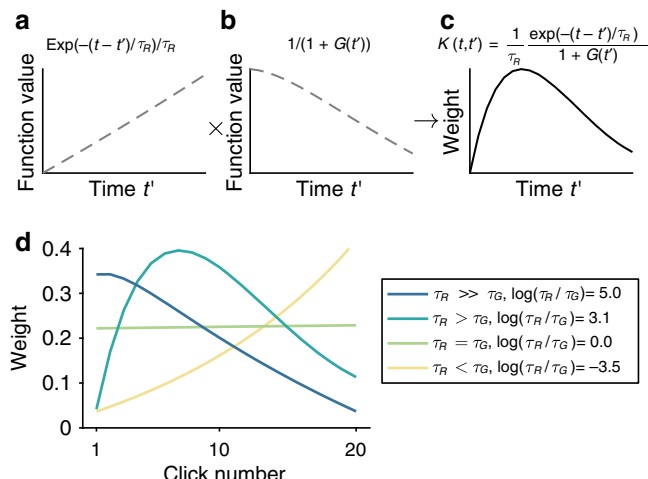

**Fig. 2 How divisive normalization generates different integration kernel shapes. a–c** Example simulation demonstrates how the two components in the integration kernel $K$ (Eq. (7)) combine to generate a bump shaped kernel. $K$ (**c**) is a product of an increasing exponential function (**a**) and the inverse of $1 + G$ (**b**), which is decreasing over time. **d** Simulations of primacy, bump, flat, and recency integration kernels using decreasing log ratios of $\tau_R$ and $\tau_G$ to demonstrate that the shape of the integration kernel is determined by a balance between the rate of the leaky integration in $R$ and the rate of the $G$ inhibition.

To investigate how integration kernels can change depending on a ratio between the rate of $R$ and the rate of $G$, we simulated the integration kernel using different $\tau_R/\tau_G$ ratios, and show that integration kernel shape changes from primacy, to bump, to flat, and then to recency as $\tau_R/\tau_G$ decreases (Fig. 2d). When $\tau_R/\tau_G$ is much larger than 1, the rate of integration is much slower than the rate of inhibition by $G$. This inhibition suppresses input from later evidence, thus producing a primacy kernel. As $\tau_R/\tau_G$ decreases towards one—$\tau_R$ decreases and $\tau_G$ increases, inhibition slows down and allows for leaky integration to happen, thus producing a bump kernel. When $\tau_R/\tau_G$ reaches one, i.e., the two rates balance out, a flat kernel is generated. Finally, when $\tau_R/\tau_G$ decreases to below one, leaky integration overcomes inhibition, generating a recency kernel.

**Humans exhibit uneven integration kernel shapes.** To examine the model in the context of behavior, we looked at behavioral data from 133 human participants. Most of these data (108 subjects) was previously published[14]. We observed that a large cohort of human participants weighed evidence unevenly when performing an auditory decision making task adapted from Poisson Clicks Task[10]. In this task, on every trial participants listened to a train of 20 clicks over 1 s at 20 Hz (Fig. 3a). Each click was on either the left or the right side. At the end of the train of clicks participants decided which side had more clicks. Participants performed between 666 and 938 trials (mean 750.8) over the course of approximately 1 h. Basic behavior in this task was comparable to that in similar perceptual decision making tasks in previous studies[10,11]. Choice exhibited a characteristic sigmoidal dependence on the net difference in clicks between left and right (Fig. 3b).

We quantified the integration kernel, i.e., the impact of every click on choice, with logistic regression in which the probability of choosing left on trial $k$ was given by

$$\text{logit}(p_{\text{left}} \text{ at trial } k) = \sum_{i=1}^{20} \beta_i^{\text{click}} \Delta C_i + \beta^{\text{bias}} \tag{8}$$

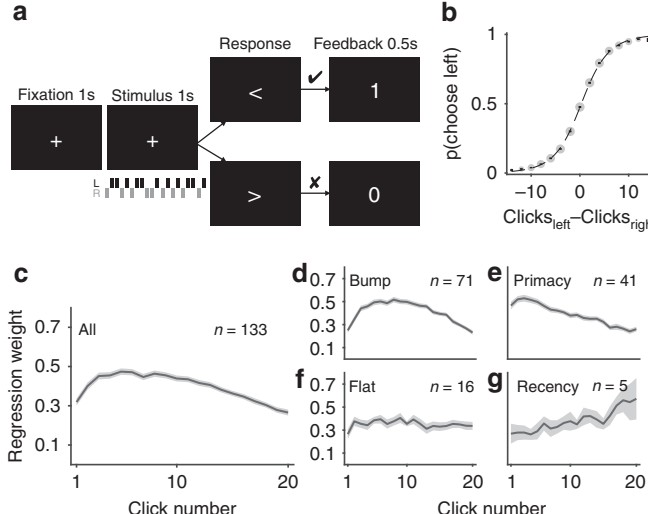

**Fig. 3 Humans exhibit uneven integration kernels in a perceptual decision making task. a** Participants listened to a train of 20 clicks coming in either the left (L, black bars) or right (R, gray bars) ear for 1s, and decided which side had more clicks. **b** Psychometric curve—choice probability (probability of choosing left)—showed sigmoidal relationship with difficulty (the difference in number of clicks between left and right). Error bars indicate s.e.m. across participants. Size of gray dots is proportional to number of trials. Dotted line indicates sigmoidal function fit. Shaded area indicates s.e.m. across participants. **c** Integration kernel, as $\beta_i^{click}$s, estimated from logistic regression (Eq. (8)), averaged across all participants. **d–g** Plots of participants' integration kernels grouped into four groups of different integration kernel shapes. All shaded areas indicate s.e.m. across participants.

where $\Delta C_i$ is the difference between left and right for the *i*th click (i.e., $\Delta C_i = \Delta C_{left,i} - \Delta C_{right,i}$, therefore, $\Delta C_i$ was +1 for a left click and −1 for right). The integration kernel was quantified by the regression weights $\beta_i^{click}$, and $\beta^{base}$ characterized the overall bias.

We found that participants weighed the clicks unevenly over time (repeated measures ANOVA on $\beta_i^{click}$: $F(19, 2508) = 34.47$, $p < 0.00001$). Importantly, post-hoc Tukey's test showed that the middle of the kernel was significantly higher than either the beginning or the end of the click (3rd–9th clicks were higher than the 1st click, and 10th–12th clicks were higher than 16th–20th clicks, $p < 0.00001$), which indicated that on average participants tended to weigh the middle of the click train more than the beginning or the end, forming a bump shaped kernel (Fig. 3c). This uneven kernel shape contributed as a source of approximately 27% of the total errors in participants' choices (see Supplementary Note 1 and Supplementary Fig. 1).

To explore individual differences in integration kernels, we further quantified the shape of the integration kernel for each participant (for detailed description of categorization of integration kernels into shapes, see Supplementary Note 2 and Supplementary Fig. 2). Specifically, we found that participants exhibited one of four distinct kernels: bump ($n = 71$, 53%), primacy ($n = 41$, 31%), flat ($n = 16$, 12%), and recency ($n = 5$, 4%) (Fig. 3d–g).

**Divisive normalization fits different kernels in humans**. To investigate whether our divisive model could account for the range of integration kernels observed in human behavior, we fit the model to participants' choices using a maximum likelihood approach. To fit the model to human behavior we assumed that a choice is made by comparing the activity in the two R units (i.e.,

$\delta = R_{left} - R_{right}$) with some noise, parameterized by $\sigma$, and an overall side bias (i.e., overall bias to either left or right). We also added an additional offset parameter $\mu$ to the kernel. With Eq. (7), the probability of choosing left at trial $k$ is given by

$$\text{logit}(p_{left} \text{ at trial } k) = \delta'(t)/\sigma + \text{bias, where } \delta'(t)$$
$$= \int_0^t (K(t,t') + \mu)\Delta C(t')dt' \tag{9}$$

We computed the probability of a choice on a given trial at $t = T$, where $T$ is the time at the end of the stimulus. The model has a total of six free parameters ($\tau_R$, $\tau_G$, inhibition weight $\omega_I$, noise $\sigma$, offset $\mu$, and an overall bias). Using parameters that best fitted to each participant's choice, we first reconstructed integration kernel from divisive normalization for each participant from the kernel function (Eqs. (7) and (9)). Divisive normalization can account for all four types of integration kernel in human participants (Fig. 4a–d and Supplementary Fig. 3). We also used divisive normalization to generate simulated choices for each participant for each trial using the best fitting parameters, and showed that the resulting psychometric curve also matched well to that of human participants (Fig. 4e and Supplementary Fig. 4). The distributions of best fitted parameters are plotted in Supplementary Figs. 5 and 6.

Our simulations in the previous section suggested that by shifting the balance between the integration and inhibition time constants (the ratio $\tau_R/\tau_G$), divisive normalization can generate the four types of kernel. We therefore examined the fitted parameter values in terms of $\tau_R/\tau_G$. We found that $\log(\tau_R/\tau_G)$ is significantly different across kernel shapes (one-way ANOVA $F(3, 129) = 12.64$, $p < 0.001$), post-hoc Tukey's test showed that this difference is driven by the difference in $\log(\tau_R/\tau_G)$ between participants with primacy kernel and bump kernel (Fig. 4f), which is in line with our prediction. However, $\log(\tau_R/\tau_G)$ in participants with flat and recency kernels are indistinguishable from either participants with either bump or primacy kernel, suggesting that the ratio of time constants is not the only factor in determining integration kernel shapes.

**Divisive normalization performs as well as DDM does**. Finally, to demonstrate that divisive normalization is comparable to an established model for such evidence accumulation tasks, we compared our model quantitatively to Drift Diffusion Model (DDM)—the predominant model used to account for this type of perceptual evidence accumulation behavior.

In its simplest form, the DDM assumes that an accumulator integrates incoming evidence over time (for example in our task the evidence +1 for a left click and −1 for a right click), with some amount of noise $\sigma_a$ added at every time step. In addition, a bias term is added to describe an overall bias to choosing either left or right. In an interrogation paradigm such as ours, a decision is made by comparing the accumulator activity with the bias when the stimulus ends, e.g., in our task, if the accumulator activity is larger than the bias, the model chooses left.

Intuitively, this form of DDM:

$$da = C(t)dt + \sigma dW \tag{10}$$

with only drift (input, i.e., clicks $C$) and diffusion (noise added by Wiener process $W$), and without any bound, would predict that every piece of evidence over time is integrated with equal weight—i.e., a flat integration kernel. Thus, the most basic form of DDM should not be able to generate a bump shaped integration kernel.

An extension can be added to the standard DDM in the form of a "memory drift" to account for primacy or recency integration kernels as well. This memory drift $\lambda$ arises out of

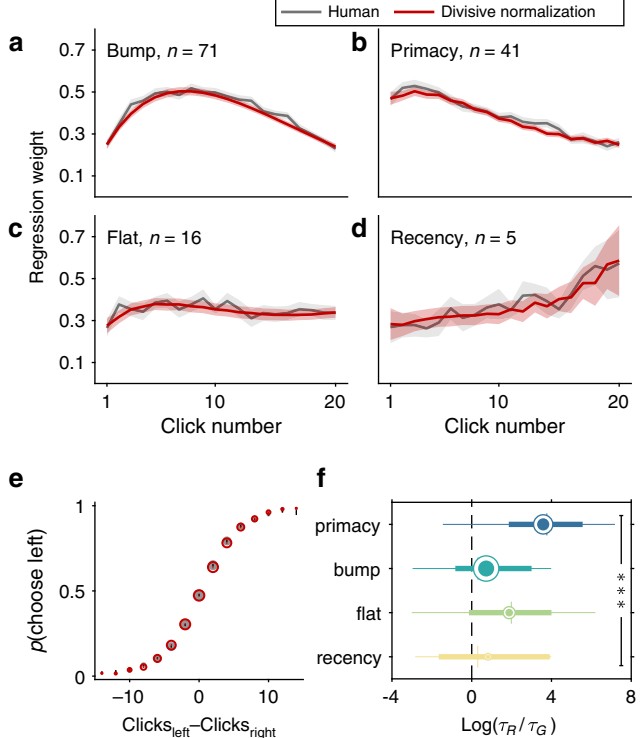

**Table 1 Divisive normalization performs as well as DDM does in formal model comparison.**

| Model | Log likelihood | AIC | BIC |
|---|---|---|---|
| Divisive normalization (6 param) | −357.7 | 727.4 | 755.1 |
| Brunton et al. DDM (6 param)[10] | −358.5 | 729.5 | 756.8 |
| LCA[15,22] | −364.2 | 738.5 | 761.6 |

**Fig. 4 Divisive normalization accounts for human integration kernels and choice curves. a–d** Integration kernels generated by divisive normalization model fitted to human participants' choices, compared to human integration kernels. Plots are grouped into groups of four different integration kernel shapes. Gray line indicates human integration kernels. Red line indicates model generated kernels. All shaded areas indicate s.e.m. across participants. **e** Psychometric curves generated by divisive normalization model compared to human psychometric curve. Gray circle indicates human psychometric curve. Red circle indicates model generated psychometric curve. Error bars indicate s.e.m. across participants. **f** Box plots of log ratios of fitted $\tau_R$ and $\tau_G$ values averaged within each integration kernel shape group. Log ratios of fitted $\tau_R$ and $\tau_G$ change significantly across kernel shape groups (***: one-way ANOVA $F(3, 129) = 12.64$, $p < 0.001$). Post-hoc Tukey's test indicates primacy kernel has a significantly higher $\tau_R$ to $\tau_G$ ratio than bump kernel has. On each box, the round marker indicates the mean and the vertical line indicates the median. The left and right edges of the box indicate the 25th and 75th percentiles, respectively. The whiskers extend to the most extreme data points not considered outliers. Size of round markers is scaled by number of participants in each kernel group.

Leaky Competitive Accumulators (LCA) model under certain constraints[15,16]:

$$da = (\lambda a + C(t))dt + \sigma dW \qquad (11)$$

$\lambda$ acts to maintain the memory of the evidence estimate. When memory is subtractive ($\lambda < 0$), DDM becomes leaky and earlier evidence is "forgotten" and thus weighed less, creating a recency bias. When memory is additive ($\lambda > 0$), accumulator activity drifts exponentially over time, and the direction of the drift is determined by the initial stimulus, thus creating a primacy effect. When $\lambda = 0$, LCA (Eq. (11)) reduces to basic DDM (Eq. (10)), and the integration kernel is flat. However, LCA alone should not be able to generate a bump shaped kernel.

Brunton and colleagues extended the LCA to include additional processes[10]: First, a bound, $B$, that describes the threshold of evidence at which the model makes a decision. In the

context of an interrogation paradigm, evidence coming after the bound has been crossed is ignored. Second, a sensory adaptation process which controls the impact of successive clicks on the same side. This process is controlled by two adaptation parameters: (1) the direction of adaptation $\phi$, which dictates whether the impact of a click on one side either increases ($\phi > 1$) or decreases ($\phi < 1$) with the number of clicks that were previously on the same side; and (2) a time constant $\tau_\phi$, that determines how quickly the adapted impact recovers to 1.

Overall the Brunton model has six free parameters—neuronal noise, memory drift, bound, two parameters controlling sensory adaptation, and bias. We fit these parameters using the maximum likelihood procedure described in the work of Brunton and colleagues[10], and following code from Yartsev and colleagues[17]. We generated choices for each participant using the best fitting parameters, and computed an integration kernel for each participant using these model generated choices.

To establish the validity of divisive normalization, we compare it with the two variations of DDM: (1) LCA and (2) Brunton DDM.

We first show that, confirming our intuition, the leak and competition in LCA (Eq. (11)) can produce a primacy, recency, and flat effect, but LCA alone cannot fit to the bump kernel (Supplementary Note 3, Supplementary Fig. 7, and Supplementary Table 1).

We found that only after introducing both bound and sensory adaptation (i.e., Brunton model) can DDM account for the behavioral data as well as divisive normalization can, both in formal model comparison using log likelihood, Akaike information criterion (AIC), and Bayesian information criterion (BIC) (Table 1), and in integration kernel and choice curve (Supplementary Note 4, Supplementary Fig. 8, and Supplementary Table 1; distribution of fitted parameters plotted in Supplementary Fig. 11).

Importantly, we show that the full Brunton DDM as reported in ref. [10] with nine parameters accounts for the behavioral data equally well (Supplementary Note 5, Supplementary Fig. 9, and Supplementary Table 1), suggesting that increasing the number of parameters did not improve model performance significantly. We also show that the LCA with the addition of just a bound does not account for the bump shaped integration kernel either (Supplementary Note 6, Supplementary Fig. 10, and Supplementary Table 1), suggesting that decreasing the number of parameters worsens the model performance. This result that divisive normalization can account for behavior as well as DDM can further support divisive normalization as a model for evidence accumulation.

## Discussion

In this work, we propose dynamic divisive normalization as a model for perceptual evidence accumulation. Theoretically, we provide a formal expression for the integration kernel, i.e., how this model weighs information over time, and we show how the shape of integration kernel falls naturally out of divisive normalization as the result of a competition between a leak term and a dynamic change in input gain. Experimentally, we show how

dynamic divisive normalization can account for the integration kernels of human participants in an auditory perceptual decision making task. In addition, with quantitative model comparison, we show that dynamic divisive normalization explains participants' choices as well as the state-of-the-art Drift Diffusion Model (DDM), the predominant model for such perceptual evidence accumulation tasks. Together, these results suggest that evidence accumulation can arise from a divisive normalization computation achieved through the interactions within a local circuit.

The result that LCA alone does not account for the bump shaped kernel is particularly interesting. Both divisive normalization and LCA produce different integration kernels via a trade-off between leak and competition. The main difference is that the competition in LCA is subtractive and the competition in divisive normalization is divisive. Superficially these two types of competition may seem similar in the sense that they both reduce accumulator activity, but they actually produce qualitatively different behavioral hypotheses. Specifically, behaviorally, we have shown that the leak and competition tradeoff in LCA alone cannot produce the bump kernel. The addition of both a bound and sensory adaptation to pure LCA is necessary to account for our human behavioral data, whereas the leak and competition trade off alone in divisive normalization can account for all four integration kernels, including the bump kernel. Of course, importantly, our results also indicate that the leak and inhibition in divisive normalization is not the only factor influencing the integration kernel shape. An interesting line of future work would be to understand how different parameters in divisive normalization trade off with each other to produce different kernel shapes.

While our findings suggest that our model accounts well for human behavior in this one task, an obvious question is whether dynamic divisive normalization is at play in other types of evidence accumulation and in other decisions? For example, the Drift Diffusion Model has been used to model evidence accumulation in a number of paradigms (from auditory clicks[10,17,18], to visual discrimination[19–21], to random dot motion[16,22–24]). Likewise, the DDM can account for choice and reaction time data in quite different settings such as memory retrieval[25], cognitive control[26], and economic and value-based decision making[27–32]. Is divisive normalization also at play in these cases? If divisive normalization is a canonical neural computation, then the simple answer is "it must be", but whether its influence extends to behavior is largely unknown (although see the emerging literature on divisive normalization in economic and value-based decisions[6,8,33]).

If people are using divisive normalization in these decisions then what computational purpose does it serve? From a computational perspective, the DDM is grounded in the sequential probability sampling test which is the optimal solution to evidence accumulation problems for two-alternative decisions under certain assumptions[16,34]. Is divisive normalization optimal under other decision making constraints? In this regard, an intriguing finding by Tajima and colleagues suggests that time-varying normalization may be almost optimal for multi-alternative decisions by implementing time-depending, nonlinear decision boundaries in a free response paradigm[35]. While such a decision boundary may not be optimal in the current task which has an interrogation paradigm, it may be optimal in a free response paradigm. This idea can be tested in future experiments with a free response paradigm where having a decision boundary is necessary.

Other advantages of divisive normalization may be its ability to encode the state of the accumulator over a wide dynamic range of evidence[1,36], or its relation to optimal Bayesian inference in some cases[37]. Of course an alternate account is that divisive normalization is necessary for other functions (e.g., balancing excitation and inhibition[1]) and the behavior we observe is simply the exhaust fumes of this function leaking out into behavior. On the other hand, stochastic choice variability is fit with a single, time-invariant noise term in divisive normalization, whereas DDM typically models noise as time (or stimulus) dependent. This does not pose a problem for the model to account for the current data, because only a single duration of the stimulus was used, but it might limit the generalizability of the model.

At the neural level, an obvious question is can our neural model explain neural data? In this regard, it is notable that our model was adapted from Louie et al.'s model of lateral intraparietal (LIP) area neurons[8]. LIP has long been thought to contain a neural representation of the state of the accumulator[38–40] and it is likely that, just like Louie's model accounts for the firing of LIP neurons in his task, our model may well be consistent with many of these past results. However, the accumulator account of LIP has recently been challenged[41–44] and other areas in prefrontal cortex[45–48] and striatum[49] have been implicated in evidence accumulation. Whether our divisive normalization explains neural firing in these areas is unknown.

On the other hand, it is important to note that the timescales in our application of the divisive normalization model should probably be interpreted as something more high-level than synaptic timescales. Normalization has long been found to be a computation that has been associated with multiple mechanisms and circuits[1]. One possible interpretation for the model in the context of our work would be that the model timescale reflects the timescale at the circuit level, and one possible reason for the variance across individuals could be due to modulation by neuromodulatory systems such as norepinephrine.

Furthermore, we note that other neural network models of evidence integration have also been proposed, perhaps most importantly the model of Wang[50]. In its simplest form, the Wang model also considers two mutually inhibiting units that, superficially, look similar to the $R$ units in Fig. 1a. However, the dynamics of the Wang model and the way it makes decisions are quite different. In particular, the mutual inhibition is calibrated in such a way that the Wang network has two stable attractor states corresponding to the outputs of the decision (e.g., left or right). The input, combined with the dynamics of the network, pushes the network into one of the two attractor states, which corresponds to the decision the network makes. Because the attraction of an attractor gets stronger the closer the network gets to it, the initial input to the model has a strong effect on the ultimate decision leading to a pronounced primacy effect in the Wang model. In contrast to Wang attractor model, our dynamic divisive normalization is essentially a line attractor network, with a single fixed point in $A$–$G$ space which is stable for all values of $\delta$ (Supplementary Note 7 and Supplementary Fig. 12). This structure allows divisive normalization to exhibit a number of different integration kernels as shown in Fig. 2 depending on the parameters. However, it is important to note that the two models are different in nature and explanatory power—specifically, the attractor model directly implements the choice mechanism, whereas the divisive normalization model implements only the decision variable computation, and choice has to be computed separately by putting the decision variable through a softmax function.

Finally, several important questions remain to be answered. The bump shaped kernel is a novel behavior in our task and stand in contrast to previously published results in humans in a similar auditory clicks task[10], that observes a flat integration kernel. So what is causing this difference? One possible explanation is that behavior in this kind of task is extremely varied across participants (as suggested by our data), and that the larger number of human participants in our study better sample the whole range of behavior. Importantly, our previous work[14] has shown that the

fitted parameter values of DDM in our participants are consistent with those reported by Brunton and colleagues[10].

In addition, Wyart and colleagues have shown that decision weights of incoming pieces of evidence fluctuated with slow cortical oscillations[51]. Even though they did not directly observe an uneven integration kernel in their data, their result that decision weights correlate with a slow rhythmic pattern is consistent with the bump kernel observed in our data, suggesting that the bump kernel may generalize to other tasks. Future work investigating the relationship between neural activity and behavior in our task would further test this idea of synchronization between cortical oscillations and integration kernel. Neural data could also shed light on why we observe such large individual differences in integration kernel (and by implication, processing time) across participants.

There is also the question of how to interpret the result that Brunton et al. model requires not only a decision bound but also sensory adaptation to account for the bump shaped kernel. It would suggest that the individual differences in kernels is caused by a difference at the sensory processing level but not at the decision making level. However, we note that the Brunton model may not be the only variant of DDM that could account for this bump shaped kernel—other potential candidates include DDM with a collapsing bound, DDM with variable drift rate, etc. Future work to answer these questions would be to compare these models on different kinds of datasets.

In sum, dynamic divisive normalization can account for human behavior in an auditory perceptual decision making task, but much evidence remains to be accumulated before we can be sure that this model is correct!

## Methods

**Participants**. One hundred eighty-eight healthy participants (University of Arizona) took part in the experiment. We analyzed the data from 133 participants (55 participants were excluded due to poor performance—accuracy lower than 60%). We have complied with all relevant ethical regulations for work with human participants. The experiment was approved by the Institutional Review Board at University of Arizona. All participants provided informed written consent prior to the experiment.

**Experimental procedures**. Participants made a series of auditory perceptual decisions. On each trial they listened to a series of 20 auditory "clicks" presented over the course of 1 s. Clicks could be either "Left" or "Right" clicks, presented in the left or right ear. Participants decided which ear received the most clicks. In contrast to the Poisson Clicks Task[10], in which the click timing was random, clicks in our task were presented every 50 ms with a fixed probability ($p = 0.55$) of occurring in the "correct" ear. The correct side was determined with a fixed 50% probability.

Participants performed the task on a desktop computer, while wearing headphones, and were positioned in chin rests to facilitate eye-tracking and pupillometry. They were instructed to fixate on a symbol displayed in the center of the screen, where response and outcome feedback was also displayed during trials, and made responses using a standard keyboard. Participants played until they made 500 correct responses or 50 min of total experiment time was reached.

**Psychometric curve**. Psychometric curves show the probability of the participant responding leftward as a function of the difference between the number of left clicks and the number of right clicks $C_{\text{left}} - C_{\text{right}}$. The identical procedure was used to produce model-predicted curves, where the model-predicted probability of choice on each trial was used instead of the participants' responses.

**Integration kernel**. To measure the contribution of each click to the participant's choice on each trial, we used logistic regression given by $\text{logit}(Y) = \beta X$, where $Y \in \{0, 1\}$ is a vector of the choice on each trial and $X$ is a matrix in which each row is the 20 clicks ($\Delta C = C_{\text{left}} - C_{\text{right}}$) on that trial, coded as +1 for left and −1 for right. The identical procedure was used to produce model-predicted integration kernels, where the model-predicted choice on each trial was used instead of the participants' responses.

**Derivation of kernel function of divisive normalization**. The model and the dynamical equations for $R$ and $G$ are described in the main text. These are

reproduced here:

$$\tau_R \frac{dR_i(t)}{dt} = -R_i(t) + \frac{C_i(t)}{1 + G(t)} \tag{1a}$$

$$\tau_G \frac{dG(t)}{dt} = -G(t) + \omega_I \sum_{i=1}^{N} R_i(t) \tag{2a}$$

From Eq. (1a) we can consider how the difference in activity $\delta(t) = R_{\text{left}}(t) - R_{\text{right}}(t)$ changes over time:

$$\tau_R \frac{d\delta(t)}{dt} = -\delta(t) + \frac{\Delta C(t)}{1 + G(t)} \tag{3a}$$

where $\Delta C(t) = C_{\text{left}}(t) - C_{\text{right}}(t)$ describes the difference in input over time.

To derive a formal expression for the kernel function, we integrate Eq. (3a) using the ansatz:

$$\delta(t) = e^{-\lambda t}\tilde{\delta}(t) \tag{4a}$$

Taking the derivative of (4a) and multiplying both sides with $\tau_R$, we get:

$$\tau_R \frac{d\delta(t)}{dt} = -\tau_R \lambda e^{-\lambda t}\tilde{\delta}(t) + \tau_R e^{-\lambda t}\frac{d\tilde{\delta}(t)}{dt} \tag{5a}$$

Combining Eqs. (3a), (4a), and (5a), we get:

$$\lambda = 1/\tau_R \tag{6a}$$

$$\tau_R e^{-\lambda t}\frac{d\tilde{\delta}(t)}{dt} = \frac{\Delta C(t)}{1 + G(t)} \tag{7a}$$

Integrating Eq. (7a) we get:

$$\tilde{\delta}(T) = \frac{1}{\tau_R}\int_0^T \frac{e^{\lambda t}\Delta C(t)}{1 + G(t)}\,\mathrm{d}t \tag{8a}$$

Substituting Eq. (8a) back into Eq. (4a), we get

$$\delta(T) = \frac{1}{\tau_R}\int_0^T \frac{\exp(-(T-t)/\tau_R)}{1 + G(t)}\Delta C(t)\,dt \tag{9a}$$

**Maximum likelihood estimate**. We fit all the models to participants' choice data using a maximum likelihood approach.

To evaluate how well a particular set of parameter values fits the behavioral data of a particular participant, we compute the probability of observing the data given the model.

Assuming the trials are independent, we can compute the probability of observing the data, **D**, given the model, $m$, as the following:

$$p(\boldsymbol{D}|\theta_m, m) = \prod_k p(d_k|\theta_m, m) \tag{10a}$$

where **D** is the full set of the participant's choices across all trials, $\theta_m$ is the set of parameters for a particular model $m$ (e.g., divisive normalization), and $d_k$ is the participant's choice on trial $k$.

The best-fit parameter values (i.e., maximum likelihood values) are the parameters $\theta_m$ that maximize the logarithm of Eq. (10a), i.e. the log likelihood LL:

$$LL = \log p(\boldsymbol{D}|\theta_m, m) = \sum_k \log p(d_k|\theta_m, m) \tag{11a}$$

where $p(d_k|\theta, m)$ is the probability of the single choice made at trial $k$ given the parameters $\theta_m$. For our model, this probability is given in Eq. (9).

**Optimization of model parameters**. After computing the log likelihood $LL$ per the description above, we then pass the negative log likelihood (whose minimum is at the same parameter values as the maximum of the positive log likelihood) to the fmincon.m function from Matlab's optimization toolbox using its interior-point algorithm, which implemented the parameter optimization. The output from fmincon.m is the parameter values that maximize the likelihood of the data.

We used on average 360 starting points (with random initial conditions) for each participant to avoid fmincon finding only the local minima and not the global minima.

The log likelihoods reported in Table 1 are the averages over all participants.

**Reporting summary**. Further information on research design is available in the Nature Research Reporting Summary linked to this article.

## Data availability

All data associated with this study are available on https://osf.io/fekpn/. A reporting summary for this Article is available as a Supplementary Information file.

## Code availability

Experiment code was created with Psychtoolbox-3 and custom MATLAB code. All analyses were created using custom MATLAB and R code. Code is available from the

corresponding author upon request, and will be uploaded to https://github.com/janekeung129/DivisiveNormModel2020. Code for fitting Brunton et al. DDM is provided by Yartsev et al.[17] (http://github.com/misun6312/PBupsModel.jl).

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

## Acknowledgements

The authors thank Samuel Feng and Alex Piet for their advice and discussion; Maxwell Alberhasky, Chrysta Andrade, Daniel Carrera, Kathryn Chung, Michael de Leon, Zamigul Dzhalilova, Asha Esprit, Abigail Foley, Emily Giron, Brittney Gonzalez, Anthony Haddad, Leah Hall, Maura Higgs, Marcus Jacobs, Min-Hee Kang, Kathryn Kellohen, Neha Kwatra, Hannah Kyllo, Alex Lawwill, Stephanie Low, Colin Lynch, Alondra Ornelas, Genevieve Patterson, Filipa Santos, Shlishaa Savita, Catie Sikora, Vera Thornton, Guillermo Vargas, Christina West, and Callie Wong for help in running the experiments.

## Author contributions

W.K. analyzed the data. W.K. and R.C.W. did the modeling work, interpreted the results, and wrote the manuscript. T.A.H. collected the data. T.A.H. and R.C.W. designed the experiment.

## Competing interests

The authors declare no competing interests.
