## [Peer Review File · Nature Communications]

Reviewers' comments:

Reviewer #1 (Remarks to the Author):

In this paper, Keung et al propose and test a divisive normalization-based model of evidence accumulation in perceptual decision making. In previous recently published results, the authors showed that human subjects integrate evidence in a time-varying manner, with a relative underweighting of early and late evidence (a bump kernel). Here, the authors show that a dynamic normalization model can capture not only a bump kernel, but other integration kernels (primacy, flat, recency), depending on the relative timescales of excitatory and inhibitory activity. In terms of theory, the authors derive a formal (but no closed-form) solution for how the decision variable under normalization evolves over time. Finally, the authors show that the dynamic normalization model predicts behavior as well as the (expanded) DDM model.

This is an intriguing and well written paper that offers a different computational explanation for dynamic decision processes in evidence integration and perceptual decision making. Given the proposed canonical nature of the computation, linking normalization to the large body of work on evidence integration is an important and interesting topic of study. I recommend accepting this paper, but do have a number of issues that I would like the authors to address and clarify first.

Major points

- (1) Clarification of the model. A few questions about the model itself. First, it's not entirely clear how click-related inputs are characterized by the model. Are they modeled as step functions in input? if so, how long are the durations? Second, the authors frame their model results almost entirely in terms of the ratio of R and G timescales; do the timescales for the different kernels depend at all of the absolute values of either of the timescales? Third, it seems that there may be a tradeoff between the G timescale and the weights w in terms of fitting particular kernels; can the authors address whether this was the case, and/or if there were any model identifiability issues?
- (2) Derivation of the normalized difference equation (Eqn 8). The description of the ansatz (Eqn 7) used to derive Eqn 8 can be more fully fleshed out. What is the importance/implication of using this particular form of the ansatz? If it's merely for mathematical tractability for the integration, it might be better to leave the ansatz out of the main text, since it is potentially confusing. If there's a particular importance of the form, then the authors should mention it in the main text.
- (3) Integration kernel in human subjects. Previous work by the authors (Nat Hum Behav, 2019) showed that most subjects also had significant past trial effects (reward and choice kernels) and side biases. However, here the authors use a model with only an integration kernel and bias. Can the authors justify why the past trial effects were left out? Presumably, history effects would not affect the shape of the intra-trial integration dynamics, but might affect overall choice prediction.
- (4) Formal model comparison. In Section 2.5, the authors compare the normalization and DDM models. Can the authors clarify exactly what metric was being used to calculate log-likelihood? Was it average choice over different stim click differences?
- (5) Interpretation of model timescales. The authors show nicely that different timescale ratios generate different integration kernels. Can the authors provide some interpretation of what this might mean at a biological level, across different subjects? In the original Louie model, these timescales were interpreted as intrinsic dynamics in the activity of excitatory and inhibitory neurons/circuits. What might this mean that different subjects have very different relative timescales? Should these timescales be interpreted as something more high-level than synaptic/biophysical timescales, which presumably don't vary much across individuals?
- (6) Comparison to the other models. There are two relevant points that could be emphasized in the Discussion, both about the ability of other models to capture different behavioral kernels. First, I think it would be helpful for a reader if the authors more explicitly discussed what a DDM model can and cannot capture in integration kernels - a summary of the DDM modeling in the Results and the Supplementary Material. For example, why does a standard model not generate a bump

kernel, and what allows the Brunton model to do so? I realize that the answer is in the sticky bound and the sensory adaptation parameter (and addressed in the SM), but for sake of comparison it is important to compare with the normalization model, which captures bump dynamics without these additional components. If anything, I think the authors could make a stronger point that the dynamic normalization model outperforms DDM variants without additional modifications.

Second, in reference to attractor models (e.g. Wang), the authors rightly point out that such models tend to overweight early information due to the nature of attractor dynamics. However, the nature and explanatory power of the models are different - in particular, the attractor model also directly implements the choice mechanism, whereas the normalization model really only implements the decision variable computation, with choice implemented by a softmax. This is an important distinction that should be explicitly mentioned.

Minor comments

(1) pg 7, line 116 - typo, "we can consider integration kernel" 
(2) pg 8, line 126 - typo, "the the inverse"

(3) pg 13, line 222 - typo, should read "same side"

Reviewer #2 (Remarks to the Author):

In this paper, the authors fit models to a previously published dataset that disclosed the time-varying weighting of evidence in humans performing an auditory integration task. They show that a divisive normalisation model that trades off leak and competition can account for the variable shapes of the integration kernel (primacy, recency, bump or flat).

Here's what I like about this paper: the authors have collected a large quantity of data from a well-designed task [albeit published previously], and have constructed a biologically plausible model to account for an interesting qualitative feature of the data. They have compared to at least one competitor (see below). The data quality are very high; the theoretical intuition in the paper is elegant (if related to previous accounts - see below). The fitting is technically accomplished and the paper is clear and well-written.

However, my overall impression is that this paper is a bit of a footnote, and perhaps better suited to a more specialised journal. Here's why:

1/ the model is only really compared to one serious rival (the 6-parameter Brunton model), and it seems to fit just as well. So there is no real way to tell the models apart from the current report alone.

2/ the specific implementation of this model to this dataset is perhaps unique, but the principle that leak and competition trade off to produce primacy and recency is well established. Its most prominent expression is in the leaky competing accumulator (LCA); see specifically Tsetsos et al 2012, *Frontiers in Neuroscience* for a detailed exploration of this phenomenon. I agree that the authors' approach here is not identical to that used previously but it seems conceptually very similar, so whilst the current report is a useful addition to the literature it seems like a relatively incremental advance.

3/ the major debate in this area is whether an any primacy might be due to the implicit bound crossing that occurs under the DDM. It was not apparent to me whether the authors considered an otherwise model with (leak and) an implicit bound; I personally do not favour this account but others do, and any contribution to this area should probably acknowledge that.

4/ The inter-individual variability in the integration kernel is very interesting, but this paper doesn't really suggest where it comes from – why do people have different kernels? Is there a normative framework that can explain the trade off between leak and competition? I would love to understand the source of this variance, and how it relates to performance in other settings, but this is not explored here.

In short, whilst I admire the authors' work, I wondered whether they might consider digging around a bit deeper in their valuable dataset to see whether there is something more novel to say about how humans perform this task.

Reviewer #3 (Remarks to the Author):

Keung, Hagen, and Wilson report an application of a dynamic divisive normalization model in an auditory perceptual decision-making task. They show that a network model with competing accumulators that are subjected to mutual and self inhibition through a dynamic divisive normalization processes can account for human performance on a click counting task, in terms of both psychometric functions and the shape of a psychophysical kernel as estimated using logistic regression. Model comparisons showed that the dynamic divisive normalization model provided a better fit to human behavior than a leaky competing accumulator model and fit equally well as an elaborated version of the drift diffusion model by Brunton and colleagues that was introduced in the context of a similar click rate discrimination task.

The manuscript makes a valuable contribution by connecting divisive normalization and evidence accumulation, two important computational perspectives on cognitive function. Although the results are somewhat equivocal in that model comparisons did not clearly support the normalization model over and above the Brunton model, the approximate equivalence of the two models is an interesting finding. And the model clearly outperforms the leaky competing accumulator on the reported data set, an interesting result that, if anything, is under-emphasized in the manuscript. With that said, I do have a few concerns about the generalizability of the finding and its relationship to the broader perceptual decision-making literature, the biological plausibility of the implied individual differences, and some aspects of the presentation.

The model is evaluated in the context of a dataset from the Wilson lab, most of which has been presented previously in a recent paper. The experiment was similar to the "poisson clicks" task and other discrete counting/rate discrimination tasks. But the task is unique in some ways — there was a fixed number of total clicks, which arrived at fixed intervals. And the experiment more broadly diverged from standard practice in psychophysics, as a large number of subjects each contributed a small-to-moderate amount of data (relative to most psychophysics experiments), with apparently little training. The large majority of subjects exhibited a distinct psychophysical kernel shape that has not really been seen in other perceptual decision making tasks (the "bump" kernel). The ability of the dynamic divisive normalization model to fit this "bump" is a key part of its support in this manuscript. But the idiosyncrasies raise some cause for concern about the generalizability of the finding, in two different ways.

One interpretation of "generalizability" is whether the model is over-fit. It is impressive how the model kernels very closely match a wide range of estimated subject kernels (Fig S3). But it is almost too impressive, given that a fairly complex and flexible model is being fit to a moderate amount of individual subject data. This concern could be alleviated if the model were fit and evaluated under cross-validation.

Ideally, it would be shown that the model fit to the first half of each subject's dataset could explain behavior on the second half. This would help address a related concern about the interpretability of quantitative analyses in the context of an experiment where subjects had relatively little experience with the task. My expectation would have been that such subjects would exhibit variability and instability in their decision-making strategy, which would be evident in the psychophysical kernels. But the model fits imply that the subjects are essentially employing the

optimal strategy and limited only by their intrinsic time constants and stochastic choice variability. Cross-validation could help resolve this concern. And relatedly, the methods section should provide more information about the format and amount of training that the subjects experienced before collection of the reported data.

A broader interpretation of "generalizability" is whether the model tells us about the computations used in other kinds of perceptual decisions. This is relevant, again, because the main feature of the data that distinguishes the model has not been seen in other experiments. Since this dataset has been published before and the focus in the manuscript is on the novel application of dynamic divisive normalization, I think the manuscript would have been much stronger if it had applied to the model to some existing datasets with other discrimination tasks. To be clear, I do not think that doing so is strictly necessary. But I would encourage the authors to consider it, as showing that divisive normalization can capture different psychophysical kernels across a range of experiments (not just across different subjects in one experiment) would be a powerful result.

One relevant feature of the model that might pose a challenge for generalizing to other experiments should be mentioned. Stochastic choice variability is fit with a single, time-invariant noise term. In contrast, the noise in evidence accumulation is typically modeled as time (or stimulus) dependent. This does not pose a problem for the model to account for the current data, because only a single duration of the stimulus was used. But it might limit the generalizability of the model in a way that should be noted.

Finally, I would also appreciate it if the authors could comment on (or otherwise support) the biological plausibility of dynamic normalization as an explanation for the individual differences. Is it realistic to expect such dramatic variability in the intrinsic time constants of these computational processes across subjects? Perhaps I am misguided, but it seems easier to accept that large individual differences in kernel shapes are caused by variations in decision strategy rather than by the low-level factors represented in the divisive normalization model.

Minor comments:

It would be useful to know how important the sensory adaptation component of the Brunton model is to its ability to explain the "bump" kernel. Would a four-parameter DDM (with independent leak and bound, but no sensory adaptation) perform as well as the divisive normalization model in a cross-validated model comparison?

It is somewhat confusing to call the lambda parameter in the LCA/Brunton models "memory noise", as it is not a stochastic component. Perhaps this parameter should instead be called "memory leak"? (That would also make it more clear in which direction it is signed without referring to the text).

I would encourage a bit more precision/clarity about exactly what is meant by "DDM" at various points. The manuscript could be read as crediting Brunton et al. with the creation of the "drift diffusion model" (pg. 13 line 205). Beyond this specific sentence, the manuscript moves confusingly back and forth between specifically discussing the "Brunton DDM" and discussing the "DDM" more generally. Indeed, as the authors point out, the classical DDM would not fit the various kernel dynamics very well at all!

The Methods section is quite brusque, particularly as pertains the model that is the focus of the experiment. (Model methods are distributed throughout the main text, but lacking some key details). I gather that the authors plan to post their modeling code online, which I applaud. But some more details about how the models were fit and evaluated in the Methods would be helpful for others who want to understand the paper and perhaps apply the model in their own experiments.

Were responses made with fingers from the same hand, or from opposite hands? This is relevant to thinking about the plausible size of the inhibitory pool.

Was the experiment self-paced, or did each trial start independent of any input from the subject? This is relevant to thinking about the weighting of the initial clicks.

In Figure S4, why do so many subjects fall into the largest single histogram bin for the sigma parameter? Was this parameter bounded during model fitting, or is this a real result?

Figure S4 shows the ratio of the two time constants, but I believe that they were estimated separately. Could the distribution for each parameter be shown?

Finally, it would be useful to see the joint distributions of these parameters across subjects, not just their marginal distributions. Instead of showing 6 histograms, perhaps a scatterplot matrix could be shown?

**Reviewer 1**

Remarks to the Author: In this paper, Keung et al propose and test a divisive nor-
malization based model of evidence accumulation in perceptual decision making. In
previous recently published results, the authors showed that human subjects integrate
evidence in a time-varying manner, with a relative underweighting of early and late ev-
idence (a bump kernel). Here, the authors show that a dynamic normalization model
can capture not only a bump kernel, but other integration kernels (primacy, flat, re-
cency), depending on the relative timescales of excitatory and inhibitory activity. In
terms of theory, the authors derive a formal (but no closed-form) solution for how the
decision variable under normalization evolves over time. Finally, the authors show that
the dynamic normalization model predicts behavior as well as the (expanded) DDM
model. This is an intriguing and well written paper that offers a different computational
explanation for dynamic decision processes in evidence integration and perceptual
decision making. Given the proposed canonical nature of the computation, linking
normalization to the large body of work on evidence integration is an important and
interesting topic of study. I recommend accepting this paper, but do have a number of
issues that I would like the authors to address and clarify first.

We thank the reviewer for such a positive response and the thorough review which has
markedly improved the paper. We enclose the following response to address all the
major and minor points raised by the reviewer.

**Point 1.1** Clarification of the model. A few questions about the model itself. First,
it's not entirely clear how click-related inputs are characterized by the model. Are
they modeled as step functions in input? if so, how long are the durations? Second,
the authors frame their model results almost entirely in terms of the ratio of R and G
timescales; do the timescales for the different kernels depend at all of the absolute val-
ues of either of the timescales? Third, it seems that there may be a tradeoff between
the G timescale and the weights w in terms of fitting particular kernels; can the au-
thors address whether this was the case, and/or if there were any model identifiability
issues?

**Response 1.1:** We thank the reviewer for pointing these details.

(1) The click-related inputs are characterized as step functions with 0.05 s duration.

(2 and 3) The absolute values of τ_R and τ_G influence the shape of the kernel in the
sense that outside certain parameter ranges the ratio doesn't predict the kernel shape
anymore. For example, if both the absolute values of τ_R and τ_G are extremely large,
the kernel will be flat regardless of the ratio between τ_R and τ_G , because the timescale
of leak and inhibition will be much longer than the timescale of the stimuli. But we
consider this situation to be relatively biologically implausible and therefore these situ-
ations are not considered in this paper.

As shown below, to a first approximation the three parameters τ_R , τ_G and ω_I tradeoff
against one another. However, this tradeoff does not hold in the general case.

**First order approximation exhibits tradeoff between τ_R , τ_G and ω_I**

To have a better understanding of trade-offs between parameters, we considered short
time approximations of the network (equations reproduced below):

$$\tau_R \frac{dR_i}{dt} = -R_i + \frac{C_i}{1 + G} \quad (1)$$

$$\tau_G \frac{dG}{dt} = -G + \omega_I \sum_{i=1}^N R_i \quad (2)$$

Let $A = R_{\text{left}} + R_{\text{right}}$ be the sum activity of R_{left} and R_{right} , we can write equations (1)

and (2) as:

$$\tau_R \frac{dA}{dt} = -A + \frac{C}{1+G} \quad (3)$$

$$\tau_G \frac{dG}{dt} = -G + \omega_I A \quad (4)$$

where C is the sum of input from the left and the right at a given time step. Since in
 our model setup, at every time step the input is set to be one click (from either left or
 right side), C can be considered here as a constant in the network.

We first look at the first-order short time approximation of the network. Assuming initial
 conditions set to zero ($A = 0, G = 0$), we can derive analytical solution of the network
 around zero initial conditions. Substituting $A = 0$ and $G = 0$ we can write equation (3)
 as:

$$\frac{dA}{dt} = \frac{C}{\tau_R} \quad (5)$$

Integrating the above equation, we get:

$$A = \frac{C}{\tau_R} t \quad (6)$$

Substituting equation (6) and $G = 0$ into equation (4), we get:

$$\frac{dG}{dt} = \frac{\omega_I C}{\tau_G \tau_R} t \quad (7)$$

Integrating the above equation gives:

$$G = \frac{\omega_I C}{\tau_G \tau_R} t^2 \quad (8)$$

Since the integration kernel depends on G (functional form of kernel reproduced be-
 low):

$$K(t, t') = \frac{1}{\tau_R} \frac{\exp(-(t - t')/\tau_R)}{1 + G(t')} \quad (9)$$

and from equation (8) we see that there is a trade-off between ω and τ_G , this trade-off
 exists in at least the first order approximation of the kernel function.

Second order approximation has no tradeoff between parameters

We then evolve the network by another time step. To get the expression of A , we sub-
 stitute the expressions of A and G we get from the first order approximation (equations
 (6) and (8)) back into equation (3):

$$\tau_R \frac{dA}{dt} = -\frac{C}{\tau_R} t + \frac{C}{1 + \frac{\omega_I C}{\tau_G \tau_R} t^2} \quad (10)$$

Integrating the above equation, we get:

$$A = -\frac{2C}{\tau_R^2} t^2 + \sqrt{\frac{\tau_G C}{\omega \tau_R}} \arctan\left(\sqrt{\frac{\omega C}{\tau_G \tau_R}} t\right) \quad (11)$$

Substituting equation (8) and (11) into equation (2), we get:

$$\frac{dG}{dt} = -\frac{\omega_I C}{\tau_G^2 \tau_R} t^2 - \frac{\omega C}{\tau_G \tau_R^2} t^2 + \sqrt{\frac{\omega C}{\tau_G \tau_R}} \arctan\left(\sqrt{\frac{\omega C}{\tau_G \tau_R}} t\right) \quad (12)$$

If we integrate the above equation, the first term will contain ω_I/τ_G^2 and the second
 term will contain ω/τ_G , indicating that ω and τ_G do not completely trade off with each
 other.

**Point 1.2** Derivation of the normalized difference equation (Eqn 8). The description
of the ansatz (Eqn 7) used to derive Eqn 8 can be more fully fleshed out. What is
the importance/implication of using this particular form of the ansatz? If it's merely for
mathematical tractability for the integration, it might be better to leave the ansatz out
of the main text, since it is potentially confusing. If there's a particular importance of
the form, then the authors should mention it in the main text.

**Response 1.2:** We thank the reviewer for this suggestion. The main reason we used
this particular form of the ansatz is indeed for mathematical tractability for the inte-
gration. Also, this ansatz would be the standard ansatz for the case where G is not
time varying (i.e. when the system is just a linear leaky integrator). We have moved it
exclusively to the Methods section.

**Point 1.3** Integration kernel in human subjects. Previous work by the authors (Nat
Hum Behav, 2019) showed that most subjects also had significant past trial effects
(reward and choice kernels) and side biases. However, here the authors use a model
with only an integration kernel and bias. Can the authors justify why the past trial
effects were left out? Presumably, history effects would not affect the shape of the
intra-trial integration dynamics, but might affect overall choice prediction.

**Response 1.3:** The main reason we did not incorporate previous trial effects is be-
cause it is hard to have a fair comparison with DDM. Especially since there has been
much debate in the field about how sequential effects are implemented in DDM - either
by drift rate or by a bias in the initial condition. In addition, the sequential effects we
found in our previous work is, while significant, quite small. So for the sake of simplicity
we omitted the sequential effects in this paper.

However, it is relatively straightforward to include sequential effect linearly in the model.

Since δ , the difference between R_{left} and R_{right} , is proportional to the logit of the proba-
bility of choosing left:

$$\text{logit}(p_{\text{left}} \text{ at trial } t) = \delta_t / \sigma + \text{bias} \quad (13)$$

we can incorporate sequential effect into the choice probability in a logistic regression
model as:

$$\text{logit}(p_{\text{left}} \text{ at trial } t) = \delta_t / \sigma + \text{bias} + \beta_{\text{previous choice}} a_{t-1} + \beta_{\text{previous correct}} a_{t-1} r_{t-1} \quad (14)$$

where a_{t-1} is the choice on the previous trial and r_{t-1} is the reward on the previous
trial (so $a_{t-1} r_{t-1}$ gives the previous correct answer).

**Point 1.4** Formal model comparison. In Section 2.5, the authors compare the normal-
ization and DDM models. Can the authors clarify exactly what metric was being used
to calculate log likelihood? Was it average choice over different stim click differences?

**Response 1.4:** The log likelihood is computed as such: for each trial, the model com-
putes a log likelihood of participant's choice. For every participant, we summed the
log likelihoods of the choices they made over all trials. The log likelihoods reported in
Table 1 are the averages over all participants. We also added the following text in our
Methods Section (page 21-22, lines 401-426).

"We fit all the models to participants' choice data using a maximum likelihood ap-
proach.

To evaluate how well a particular set of parameter values fits the behavioural data of
a particular participant, we compute the probability of observing the data given the
model.

Assuming the trials are independent, we can compute the probability of observing the
data, D , given the model, m , as the following:

$$p(D|\theta_m, m) = \prod_k p(d_k|\theta_m, m) \quad (15)$$

where D is the full set of the participant's choices across all trials, θ_m is the set of
parameters for a particular model m (e.g. divisive normalization), and d_k is the partici-
pant's choice on trial k .

The best-fit parameter values (i.e. maximum likelihood values) are the parameters θ_m
that maximize the logarithm of equation (15), i.e. the log likelihood LL :

$$LL = \log p(D|\theta_m, m) = \sum_k \log p(d_k|\theta_m, m) \quad (16)$$

where $p(d_k|\theta, m)$ is the probability of the single choice made at trial k given the param-

eters θ_m . For our model, this probability is given in equation (9) in the Main Text.

After computing the log likelihood LL per the description above, we then pass the neg-
ative log likelihood (whose minimum is at the same parameter values as the maximum
of the positive log likelihood) to the `fmincon.m` function from Matlab's optimization tool-
box using its interior-point algorithm, which implemented the parameter optimization.
The output from `fmincon.m` are the parameter values that maximize the likelihood of
the data.

We used on average 360 starting points (with random initial conditions) for each par-
ticipant to avoid `fmincon` finding only the local minima and not the global minima.

The log likelihoods reported in Main Text Table 1 are the averages over all partici-
pants.”

**Point 1.5** Interpretation of model timescales. The authors show nicely that different
timescale ratios generate different integration kernels. Can the authors provide some
interpretation of what this might mean at a biological level, across different subjects?
In the original Louie model, these timescales were interpreted as intrinsic dynamics
in the activity of excitatory and inhibitory neurons/circuits. What might this mean that
different subjects have very different relative timescales? Should these timescales be
interpreted as something more high-level than synaptic/biophysical timescales, which
presumably don't vary much across individuals?

**Response 1.5:** We agree that the timescales in our application of the divisive nor-
malization model should be interpreted as something more high-level than synaptic
timescales. Normalization has long been found to be a computation that have been
associated with multiple mechanisms and circuits (Carandini and Heeger, 2011). One
possible interpretation for the model in the context of our work would be that the model
timescale reflects the timescale at the circuit level, and one possible reason for the
variance across individuals could be due to modulation by neuromodulatory systems
such as norepinephrine.

We have added the above paragraph to Discussion (page 17, lines 323-329).

**Point 1.6** Comparison to the other models. There are two relevant points that could
be emphasized in the Discussion, both about the ability of other models to capture
different behavioural kernels. First, I think it would be helpful for a reader if the authors
more explicitly discussed what a DDM model can and cannot capture in integration
kernels - a summary of the DDM modeling in the Results and the Supplementary
Material. For example, why does a standard model not generate a bump kernel, and
what allows the Brunton model to do so? I realize that the answer is in the sticky
bound and the sensory adaptation parameter (and addressed in the SM), but for sake
of comparison it is important to compare with the normalization model, which captures
bump dynamics without these additional components. If anything, I think the authors
could make a stronger point that the dynamic normalization model outperforms DDM
variants without additional modifications.

Second, in reference to attractor models (e.g. Wang), the authors rightly point out
that such models tend to overweight early information due to the nature of attractor
dynamics. However, the nature and explanatory power of the models are different
- in particular, the attractor model also directly implements the choice mechanism,
whereas the normalization model really only implements the decision variable com-
putation, with choice implemented by a softmax. This is an important distinction that
should be explicitly mentioned.

**Response 1.6:** We thank the reviewer for these two important suggestions.

(1) We have reorganized the manuscript to highlight the difference between divisive
normalization and DDM in the Main Text (page 14, lines 244-248), and Discussion
(page 16, lines 274-284), quoted below:

Main Text (page 14, lines 244-248)

"We first show that, confirming our intuition, the leak and competition in LCA can pro-
duce a primacy, recency, and flat effect, but LCA alone cannot fit to the bump kernel.

We found that only after introducing both bound and sensory adaptation (i.e. Brunton
model) can DDM account for the behavioural data as well as divisive normalization
can.”

Discussion (page 16, lines 274-284)

”The result that LCA alone does not account for the bump shaped kernel is partic-
ularly interesting. Both divisive normalization and LCA produce different integration
kernels via a trade-off between leak and competition. The main difference is that the
competition in LCA is subtractive and the competition in divisive normalization is divi-
sive. Superficially these two types of competition may seem similar in the sense that
they both reduce accumulator activity, but they actually produce qualitatively different
behavioural hypotheses. Specifically, behaviourally, we had shown that the leak and
competition tradeoff in LCA alone cannot produce the bump kernel. The addition of
both a bound and sensory adaptation to pure LCA is necessary to account for our
human behavioural data, whereas the leak and competition trade off alone in divisive
normalization can account for all four integration kernels, including the bump kernel.”

(2) We have also added the following text to make the important distinction between
attractor model and normalization model in the Discussion (page 18, lines 346-350).

”However, it is important to note that the two models are different in nature and ex-
planatory power — specifically, the attractor model directly implements the choice
mechanism, whereas the divisive normalization model implements only the decision
variable computation, and choice has to be computed separately by putting the deci-
sion variable through a softmax function.”

**Minor points:**

(1) pg 7, line 116 - typo, "we can consider integration kernel" fixed

(2) pg 8, line 126 - typo, "the the inverse" fixed

(3) pg 13, line 222 - typo, should read "same side" fixed

**Reviewer 2**

**Remarks to the Author:** In this paper, the authors fit models to a previously published
dataset that disclosed the time-varying weighting of evidence in humans performing an
auditory integration task. They show that a divisive normalisation model that trades off
leak and competition can account for the variable shapes of the integration kernel (pri-
macy, recency, bump or flat). Here's what I like about this paper: the authors have col-
lected a large quantity of data from a well-designed task [albeit published previously],
and have constructed a biologically plausible model to account for an interesting quali-
tative feature of the data. They have compared to at least one competitor (see below).
The data quality are very high; the theoretical intuition in the paper is elegant (if re-
lated to previous accounts – see below). The fitting is technically accomplished and
the paper is clear and well-written. However, my overall impression is that this paper
is a bit of a footnote, and perhaps better suited to a more specialised journal. Here's
why:

We appreciate the positive comments and thank the reviewer for the suggestions. We
apologize for not having highlighted enough the difference between our model and
DDM/LCA. Here we address the comments made the reviewer.

**Point 2.1** The model is only really compared to one serious rival (the 6-parameter
Brunton model), and it seems to fit just as well. So there is no real way to tell the
models apart from the current report alone.

**Response 2.1:** We agree that the current data cannot distinguish between the two
models. However, we find it notable that the divisive model accounts for behavior as
well as the established DDM but with quite different underlying mechanism.

With regard to the LCA and other models, we apologize for not being more clear about
the range of models tested and the relationship between DDM and LCA. In particular,
our results show that LCA does not fit behavior as well as divisive normalization nor
can it account for the bump without the addition of a bound. (See **Response 2.2** below
for detailed description).

**Point 2.2** The specific implementation of this model to this dataset is perhaps unique,
but the principle that leak and competition trade off to produce primacy and recency is
well established. Its most prominent expression is in the leaky competing accumulator
(LCA); see specifically Tsetsos et al 2012, *Frontiers in Neuroscience* for a detailed ex-
ploration of this phenomenon. I agree that the authors' approach here is not identical to
that used previously but it seems conceptually very similar, so whilst the current report
is a useful addition to the literature it seems like a relatively incremental advance.

**Response 2.2:** While we agree that a the idea of a tradeoff between leak and compe-
tition is not unique to our model, there is a key qualitative difference between the LCA
and the divisive normalization model. Specifically, competition in LCA is subtractive
and the competition in divisive normalization is divisive. Superficially these two types
of competition may seem similar in the sense that they both reduce accumulator activ-
ity, but they actually produce qualitatively different behavioral and neural hypotheses.

Specifically, behaviorally, we had shown that the leak and competition tradeoff in LCA
alone cannot produce the bump kernel. We apologize for not having emphasized this
result, but we have now rearranged the manuscript such that this result is highlighted
(Main Text (page 14, lines 244-248), and Discussion (page 16, lines 274-284), quoted
below). The addition of both a bound and sensory adaptation to pure LCA is necessary
to account for our human behavioral data, while as the leak and competition trade off
alone in divisive normalization can account for all four integration kernels, including the
bump kernel.

Neurally, previous work has suggested that divisive and subtractive mechanisms of
competition have different neural implementations (Wilson et al. 2012 *Nature*, Ayaz
and Chance 2009 *J Neurophysiol*). This suggests that divisive normalization and LCA
produces different neural hypotheses for evidence integration.

Addition to text:

Main Text (page 14, lines 244-248)

"We first show that, confirming our intuition, the leak and competition in LCA can pro-
duce a primacy, recency, and flat effect, but LCA alone cannot fit to the bump kernel.

We found that only after introducing both bound and sensory adaptation (i.e. Brunton
model) can DDM account for the behavioural data as well as divisive normalization
can."

Discussion (page 16, lines 274-284)

"The result that LCA alone does not account for the bump shaped kernel is partic-
ularly interesting. Both divisive normalization and LCA produce different integration
kernels via a trade-off between leak and competition. The main difference is that the
competition in LCA is subtractive and the competition in divisive normalization is divi-
sive. Superficially these two types of competition may seem similar in the sense that
they both reduce accumulator activity, but they actually produce qualitatively different
behavioural hypotheses. Specifically, behaviourally, we had shown that the leak and
competition tradeoff in LCA alone cannot produce the bump kernel. The addition of
both a bound and sensory adaptation to pure LCA is necessary to account for our
human behavioural data, whereas the leak and competition trade off alone in divisive
normalization can account for all four integration kernels, including the bump kernel."

**Point 2.3** The major debate in this area is whether an any primacy might be due to
the implicit bound crossing that occurs under the DDM. It was not apparent to me
whether the authors considered an otherwise model with (leak and) an implicit bound;
I personally do not favour this account but others do, and any contribution to this area
should probably acknowledge that.

**Response 2.3:** We apologize for the confusion in our texts. The main model that we
compared divisive normalization to — the Brunton 6 parameter model — is an LCA
(with leak, noise, bias) and the addition of bound and sensory adaptation. We have
made changes in how we refer to different models in the Main Text (page 13, lines
213-228) to hopefully make this clearer.

In addition, we also find that an LCA (with leak, noise, bias) and the addition of just
a bound (i.e. no sensory adaptation) performs worse than divisive normalization. We
tested this in a cross-validated model comparison. For every participant, we fit the
models to odd numbered trials and test the fitted models on the even numbered trials
by computing the log likelihood of choices on the even numbered trials, and vice versa.
(Response Table 1). Importantly, LCA with just bound does not fit to the bump inte-
gration kernel (Response Figure 1), whereas both divisive normalization and Brunton
model fit to the bump kernel in cross-validation analysis (Response Figure 2).

model	log lik (tested on even no. trials)	log lik (tested on odd no. trials)
Div.norm.	-182.4	-184.6
bounded LCA [10]	-183.1	-185.2

Response Table 1: Divisive normalization fits to choice data as well as LCA does in cross-validation analysis. The middle column shows log likelihood values (averaged across participants) of model fitted to odd numbered trials and tested on even numbered trials. Vice versa in the right column.

Response Figure 1: Cross-validation of LCA (with bound but without sensory adaptation) on data separated by every other trial. (a) LCA fitted to odd numbered trials tested on even numbered trials. (b) LCA fitted to even numbered trials tested on odd numbered trials. Black solid and dashed lines show integration kernels of odd and even numbered trials respectively.

Response Figure 2: Cross-validation of divisive normalization and Brunton model (LCA with bound and with sensory adaptation) on data separated by every other trial. (a,c) Model fitted to odd numbered trials tested on even numbered trials. (b,d) Model fitted to even numbered trials tested on odd numbered trials. (a-b) Divisive normalization cross validation fits. (c-d) Brunton model cross validation fits. Red and blue lines show divisive normalization and Brunton model generated integration kernel. Black solid and dashed lines show integration kernels of odd and even numbered trials respectively.

**Point 2.4** The inter-individual variability in the integration kernel is very interesting, but
this paper doesn't really suggest where it comes from – why do people have different
kernels? Is there a normative framework that can explain the trade off between leak
and competition? I would love to understand the source of this variance, and how it
relates to performance in other settings, but this is not explored here. In short, whilst
I admire the authors' work, I wondered whether they might consider digging around a
bit deeper in their valuable dataset to see whether there is something more novel to
say about how humans perform this task.

**Response 2.4:** It is an interesting question. Tajima and colleagues' recent work [35]
suggests that time-dependent normalization (plus an urgency signal) implements close-
to-optimal decisions to multi-alternative choices. Specifically, they found that the op-
timal decision boundaries (in a multi-alternative choice task) are nonlinear. The time-
dependent normalization helps approximate the optimal decisions by effectively creat-
ing nonlinear decision boundaries that are similar to the optimal decision boundaries,
in contrast to the linear decision boundaries in traditional race models. In our case,
the leak and competition in the dynamic divisive normalization may be implementing
a similar nonlinear decision boundary, as the time-dependent normalization in Tajima
et al.'s model. This idea can be tested in future experiments with a free response
paradigm where having a decision boundary is necessary.

**Reviewer 3**

**Remarks to the Author:** Keung, Hagen, and Wilson report an application of a dy-
namic divisive normalization model in an auditory perceptual decision-making task.
They show that a network model with competing accumulators that are subjected to
mutual and self inhibition through a dynamic divisive normalization processes can ac-
count for human performance on a click counting task, in terms of both psychometric
functions and the shape of a psychophysical kernel as estimated using logistic re-
gression. Model comparisons showed that the dynamic divisive normalization model
provided a better fit to human behavior than a leaky competing accumulator model
and fit equally well as an elaborated version of the drift diffusion model by Brunton and
colleagues that was introduced in the context of a similar click rate discrimination task.

The manuscript makes a valuable contribution by connecting divisive normalization
and evidence accumulation, two important computational perspectives on cognitive
function. Although the results are somewhat equivocal in that model comparisons did
not clearly support the normalization model over and above the Brunton model, the
approximate equivalence of the two models is an interesting finding. And the model
clearly outperforms the leaky competing accumulator on the reported data set, an in-
teresting result that, if anything, is under-emphasized in the manuscript. With that said,
I do have a few concerns about the generalizability of the finding and its relationship
to the broader perceptual decision-making literature, the biological plausibility of the
implied individual differences, and some aspects of the presentation.

We appreciate the positive response and we thank the reviewer for the feedback which
has helped markedly improve the manuscript. For ease of reading, we break up the
reviewer's comments into points, and we address the comments point-by-point.

**Point 3.1** The model is evaluated in the context of a dataset from the Wilson lab, most
of which has been presented previously in a recent paper. The experiment was similar
to the "poisson clicks" task and other discrete counting/rate discrimination tasks. But
the task is unique in some ways — there was a fixed number of total clicks, which
arrived at fixed intervals. And the experiment more broadly diverged from standard
practice in psychophysics, as a large number of subjects each contributed a small-to-
moderate amount of data (relative to most psychophysics experiments), with appar-
ently little training. The large majority of subjects exhibited a distinct psychophysical
kernel shape that has not really been seen in other perceptual decision making tasks
(the "bump" kernel). The ability of the dynamic divisive normalization model to fit this
"bump" is a key part of its support in this manuscript. But the idiosyncrasies raise some
cause for concern about the generalizability of the finding, in two different ways.

One interpretation of "generalizability" is whether the model is over-fit. It is impressive
how the model kernels very closely match a wide range of estimated subject kernels
(Fig S3). But it is almost too impressive, given that a fairly complex and flexible model
is being fit to a moderate amount of individual subject data. This concern could be
alleviated if the model were fit and evaluated under cross-validation.

Ideally, it would be shown that the model fit to the first half of each subject's dataset
could explain behavior on the second half. This would help address a related concern
about the interpretability of quantitative analyses in the context of an experiment where
subjects had relatively little experience with the task. My expectation would have been
that such subjects would exhibit variability and instability in their decision-making strat-
egy, which would be evident in the psychophysical kernels. But the model fits imply
that the subjects are essentially employing the optimal strategy and limited only by
their intrinsic time constants and stochastic choice variability. Cross-validation could
help resolve this concern. And relatedly, the methods section should provide more in-
formation about the format and amount of training that the subjects experienced before
collection of the reported data.

A broader interpretation of "generalizability" is whether the model tells us about the
computations used in other kinds of perceptual decisions. This is relevant, again, be-
cause the main feature of the data that distinguishes the model has not been seen in
other experiments. Since this dataset has been published before and the focus in the
manuscript is on the novel application of dynamic divisive normalization, I think the
manuscript would have been much stronger if it had applied to the model to some ex-
isting datasets with other discrimination tasks. To be clear, I do not think that doing so
is strictly necessary. But I would encourage the authors to consider it, as showing that
divisive normalization can capture different psychophysical kernels across a range of
experiments (not just across different subjects in one experiment) would be a powerful
result.

**Response 3.1:** As we understand, the concern here is about the generalizability of
the model - mainly, how well the model generalizes within subject. There are two con-
cerns here: one is whether the model overfitted, and the other is whether participants'
integration kernels changed over time. Here we address each separately.

To check whether the model over-fitted to the data, we split the data into two sets
by every other trial (i.e. one set contains all the odd number trials and the other set
contains all the even number trials). We did cross-validation by fitting the model to one
set and then testing it on the other set (Response Figure 3). For quantitative analysis,
we compared divisive normalization and the Brunton model. Since we are doing cross
validation, we compare the log likelihood directly, and we find that both models fit the
choice data similarly well in cross-validated analysis (Response Table 2).

model	log lik (tested on even no. trials)	log lik (tested on odd no. trials)
Div.norm.	-182.4	-184.6
DDM [10]	-182.7	-184.5

Response Table 2: Divisive normalization fits to choice data as well as DDM does in cross-validation analysis. The middle column shows log likelihood values (averaged across participants) of model fitted to odd numbered trials and tested on even numbered trials. Vice versa in the right column.

Response Figure 3: Cross-validation of model on data separated by every other trial. (a,c) Model fitted to odd numbered trials tested on even numbered trials. (b,d) Model fitted to even numbered trials tested on odd numbered trials. (a-b) Divisive normalization cross validation fits. (c-d) Brunton model cross validation fits. Red and blue lines show divisive normalization and DDM generated integration kernel. Black solid and dashed lines show integration kernels of odd and even numbered trials respectively.

To look at whether participants' integration kernels changed over the course of the ex-
 periment, we split the data into first and second halves and compared the integration
 kernels of two halves. We found that participants' integration kernels did change over
 the course of the experiment (Response Figure 4). However, importantly, the bump
 shape of the integration kernel was preserved in both halves, suggesting that partic-
 ipants still used a predominantly bump shaped integration kernel throughout the ex-
 periment. We also showed that neither divisive normalization nor Brunton model did a
 good job at generalizing across the two halves (Response Figure 4). Again, comparing
 log likelihoods, we find that both models fit the choice data similarly in cross-validated
 analysis (Response Table 3).

model	log lik (tested on 2nd half)	log lik (tested on 1st half)
Div.norm.	-182.5	-192.1
DDM [10]	-182.3	-192.0

Response Table 3: Divisive normalization fits to choice data as well as DDM does in cross-validation analysis. The middle column shows log likelihood values (averaged across participants) of model fitted to first half on the experiment and tested on the second half. Vice versa in the right column.

Response Figure 4: Cross-validation of model on data separated by first and second halves. (a,c) Model fitted to first half of data tested on the second half. (b,d) Model fitted to second half of data tested on the first half. (a-b) Divisive normalization cross validation fits. (c-d) Brunton model cross validation fits. Red and blue lines show divisive normalization and DDM generated integration kernel. Black solid and dashed lines show integration kernels of first and second halves respectively.

**Point 3.2** One relevant feature of the model that might pose a challenge for generaliz-
ing to other experiments should be mentioned. Stochastic choice variability is fit with
a single, time-invariant noise term. In contrast, the noise in evidence accumulation is
typically modeled as time (or stimulus) dependent. This does not pose a problem for
the model to account for the current data, because only a single duration of the stim-
ulus was used. But it might limit the generalizability of the model in a way that should
be noted.

**Response 3.2:** We appreciate this suggestion and have added the following text to
the Discussion (page 17, lines 308-313).

"Stochastic choice variability is fit with a single, time-invariant noise term in divisive
normalization, whereas DDM typically models noise as time (or stimulus) dependent.
This does not pose a problem for the model to account for the current data, because
only a single duration of the stimulus was used, but it might limit the generalizability of
the model."

**Point 3.3** Finally, I would also appreciate it if the authors could comment on (or oth-
erwise support) the biological plausibility of dynamic normalization as an explanation
for the individual differences. Is it realistic to expect such dramatic variability in the
intrinsic time constants of these computational processes across subjects? Perhaps
I am misguided, but it seems easier to accept that large individual differences in ker-
nel shapes are caused by variations in decision strategy rather than by the lower-level
factors represented in the divisive normalization model.

**Response 3.3:** We thank the reviewer for raising this interesting point. We agree
that the timescales in our application of the divisive normalization model should be
interpreted as something more high-level than synaptic timescales. One possible in-
terpretation would be that the model timescale reflects the timescale at the circuit level,
and differences across subjects are modulated by neuromodulatory systems such as
norepinephrine.

We have added the above paragraph to Discussion (page 17, lines 323-329).

**Minor points:**

– It would be useful to know how important the sensory adaptation component of the
Brunton model is to its ability to explain the "bump" kernel. Would a four-parameter
DDM (with independent leak and bound, but no sensory adaptation) perform as well
as the divisive normalization model in a cross-validated model comparison?

Thank you for this suggestion. We find that a four-parameter DDM (with leak, bound,
noise, and bias) performs worse than divisive normalization in a cross-validated model
comparison (Response Table 4). Importantly, four-parameter DDM does not fit to the
bump integration kernel (Response Figure 5).

model	log lik (tested on even no. trials)	log lik (tested on odd no. trials)
Div.norm.	-182.4	-184.6
DDM (4 param) [10]	-183.1	-185.2

Response Table 4: Divisive normalization fits to choice data as well as DDM does in cross-validation analysis. The middle column shows log likelihood values (averaged across participants) of model fitted to odd numbered trials and tested on even numbered trials. Vice versa in the right column.

Response Figure 5: Cross-validation of four-parameter DDM (with bound but without adaptation) on data separated by every other trial. (a) DDM (4 param) fitted to odd numbered trials tested on even numbered trials. (b) DDM (4 param) fitted to even numbered trials tested on odd numbered trials. Black solid and dashed lines show integration kernels of odd and even numbered trials respectively.

– It is somewhat confusing to call the lambda parameter in the LCA/Brunton models
"memory noise", as it is not a stochastic component. Perhaps this parameter should in-
stead be called "memory leak"? (That would also make it more clear in which direction
it is signed without referring to the text).

We have changed it to “memory drift”, since it could either be leaky or impulsive, and
it is also what is called in the original Brunton et al. paper.

– I would encourage a bit more precision/clarity about exactly what is meant by “DDM”
at various points. The manuscript could be read as crediting Brunton et al. with the
creation of the “drift diffusion model” (pg. 13 line 205). Beyond this specific sentence,
the manuscript moves confusingly back and forth between specifically discussing the
“Brunton DDM” and discussing the “DDM” more generally. Indeed, as the authors point
out, the classical DDM would not fit the various kernel dynamics very well at all!

We thank the reviewer for this suggestion. We originally hoped to avoid confusion by
naming all the models “DDM”, but it’s clear that that has been more confusing. We
have changed to more precise names for the models (DDM, LCA, Brunton DDM, etc)
so that it is clearer. The changes can mainly be seen in Section 2.5 in the main text.

– The Methods section is quite brusque, particularly as pertains the model that is the
focus of the experiment. (Model methods are distributed throughout the main text,
but lacking some key details). I gather that the authors plan to post their modeling
code online, which I applaud. But some more details about how the models were fit
and evaluated in the Methods would be helpful for others who want to understand the
paper and perhaps apply the model in their own experiments.

We added the following additional details to model fitting in the Methods section (page
21-22, lines 401-426).

“We fit all the models to participants’ choice data using a maximum likelihood ap-
proach.

To evaluate how well a particular set of parameter values fits the behavioural data of
a particular participant, we compute the probability of observing the data given the
model.

Assuming the trials are independent, we can compute the probability of observing the

data, D , given the model, m , as the following:

$$p(D|\theta_m, m) = \prod_k p(d_k|\theta_m, m) \quad (17)$$

where D is the full set of the participant's choices across all trials, θ_m is the set of
parameters for a particular model m (e.g. divisive normalization), and d_k is the partici-
pant's choice on trial k .

The best-fit parameter values (i.e. maximum likelihood values) are the parameters θ_m
that maximize the logarithm of equation (17), i.e. the log likelihood LL :

$$LL = \log p(D|\theta_m, m) = \sum_k \log p(d_k|\theta_m, m) \quad (18)$$

where $p(d_k|\theta, m)$ is the probability of the single choice made at trial k given the param-
eters θ_m . For our model, this probability is given in equation (9) in the Main Text.

After computing the log likelihood LL per the description above, we then pass the neg-
ative log likelihood (whose minimum is at the same parameter values as the maximum
of the positive log likelihood) to the `fmincon.m` function from Matlab's optimization tool-
box using its interior-point algorithm, which implemented the parameter optimization.
The output from `fmincon.m` are the parameter values that maximize the likelihood of
the data.

We used on average 360 starting points (with random initial conditions) for each par-
ticipant to avoid `fmincon` finding only the local minima and not the global minima.

The log likelihoods reported in Main Text Table 1 are the averages over all participants."

– Were responses made with fingers from the same hand, or from opposite hands?

This is relevant to thinking about the plausible size of the inhibitory pool.

The responses were made with fingers from the same hand.

Response Figure 6: Histogram of fitted divisive normalization parameters.

– Was the experiment self-paced, or did each trial start independent of any input from
 the subject? This is relevant to thinking about the weighting of the initial clicks.

The experiment was not self paced. But we have run a version where the experiment
 was self paced. We saw the same bump shaped integration kernel in the preliminary
 data of 21 participants. This suggests that the lower weighting of the initial few clicks
 was not due to timing or a lack of preparation.

– In Figure S4, why do so many subjects fall into the largest single histogram bin for
 the sigma parameter? Was this parameter bounded during model fitting, or is this a
 real result?

Thank you for pointing this out. The reported "sigma" values are actually $1/\sigma$ instead of
 σ . We have corrected the parameter histogram (reproduced below) in Supplementary
 Section 4 Figure S5 (page 35).

– Figure S4 shows the ratio of the two time constants, but I believe that they were

estimated separately. Could the distribution for each parameter be shown?

Yes, we added the parameter distribution histograms of the two time constants (see
figure above) to Supplementary Section 4 Figure S5 (page 35).

– Finally, it would be useful to see the joint distributions of these parameters across
subjects, not just their marginal distributions. Instead of showing 6 histograms, per-
haps a scatterplot matrix could be shown?

We thank the reviewer for this suggestion, we plot the scatter plots of joint distribu-
tions of these parameters (Response Figure 7) and added to Supplementary Section
4 (page 35).

Response Figure 7: Parameter distributions of divisive normalization

****REVIEWERS' COMMENTS:**

Reviewer #1 (Remarks to the Author):

The authors have been very responsive to my previous comments, and I think their revisions - particularly regarding conceptual framing and discussion - have made the paper better suited for a general audience. This paper offers a potential alternative framework to the widely-assumed accumulation-to-bound models and should spur much future research - I recommend publication.

Reviewer #2 (Remarks to the Author):

The authors engaged very seriously with the reviews. As I see it, the major improvements are 1) that they show clearly that the "bump" effect they are interested in cannot be explained by other models, including vanilla DDM and LCA; they clarify the relationship between the Brunton model and these theories; and they explore the data in some more detail, including the temporal split analysis proposed by Rev.3.

I am happy with the replies to my comments. I would persist in a slight discomfort that the paper's main message is "here's a model that fits" rather than "here's a computational principle of interest (and here's why)". I think I might have trawled through this marvellous dataset a bit more to try to identify clear qualitative effects that are theory-constraining. But, it's not my paper, it's the authors, and so I'm happy for it to proceed as is.

Reviewer #3 (Remarks to the Author):

This is a review of a revised submission by Keung and colleagues reporting the application of a divisive normalization model to a two-choice decision-making task where auditory evidence can be accumulated over time.

The revised manuscript has improved clarity on several technical points, and I think the dynamic normalization solution for evidence accumulation is now better contrasted with approaches based on subtractive competition. The manuscript also partially addresses my concern about the generalizability of the results.

In the rebuttal letter, the authors show additional analyses using cross-validation; these largely resolve my concern over whether the models are over-fit (at least at the group level). The new analyses also suggest that, while performance was non-stationary over the duration of the experiment, this can't explain the bump kernel. Together, the new analyses suggest that the modeling results will generalize beyond this specific dataset.

On the other hand, the revised manuscript does not really address concerns about generalization in the broader sense: whether these insights will extend to other decision-making tasks or to decision-making as a cognitive process. As noted, the primary motivation for the divisive normalization model is its parsimonious explanation for the "bump" kernel exhibited by a preponderance of subjects in this task. But the bump kernel has not been reported previously in the literature (based on the supplemental plots in Brunton et al. 2013, it does not appear even in the Poisson clicks task, which would seem the closest analogue). Neither the rebuttal nor revised manuscript address this concern.

Ultimately, the best way to answer this question would be to compare these models on different kinds of datasets. That may be out of scope for this manuscript — I think it makes a useful contribution while remaining focused on the authors' previously published data. But in the absence of such a comparison, several important questions remain unanswered. Is there something special about the Bernoulli clicks task that reveals the bump kernel? If so, what? Is it meaningful that the Brunton DDM can account for it with parameters designed to model nonlinearities in sensory processing (suggesting that the bump kernel is not a feature of the decision-making process per se)? The overall sense of uncertainty is exacerbated by the lack of explanation for the striking

individual differences in integration kernels (and by implication, processing time constants). To be fair, the authors sketch a possible explanation, but one that remains rather speculative.

Despite these residual concerns, the revised manuscript is technically strong, centers on important questions in computational neuroscience, and suggests some interesting new ways to think about those questions. I expect that it will motivate future work to resolve some of the outstanding questions.

Reviewer #3 (Remarks to the Author):

This is a review of a revised submission by Keung and colleagues reporting the application of a divisive normalization model to a two-choice decision-making task where auditory evidence can be accumulated over time.

The revised manuscript has improved clarity on several technical points, and I think the dynamic normalization solution for evidence accumulation is now better contrasted with approaches based on subtractive competition. The manuscript also partially addresses my concern about the generalizability of the results.

In the rebuttal letter, the authors show additional analyses using cross-validation; these largely resolve my concern over whether the models are over-fit (at least at the group level). The new analyses also suggest that, while performance was non-stationary over the duration of the experiment, this can't explain the bump kernel. Together, the new analyses suggest that the modeling results will generalize beyond this specific dataset.

On the other hand, the revised manuscript does not really address concerns about generalization in the broader sense: whether these insights will extend to other decision-making tasks or to decision-making as a cognitive process. As noted, the primary motivation for the divisive normalization model is its parsimonious explanation for the "bump" kernel exhibited by a preponderance of subjects in this task. But the bump kernel has not been reported previously in the literature (based on the supplemental plots in Brunton et al. 2013, it does not appear even in the Poisson clicks task, which would seem the closest analogue). Neither the rebuttal nor revised manuscript address this concern.

Ultimately, the best way to answer this question would be to compare these models on different kinds of datasets. That may be out of scope for this manuscript — I think it makes a useful contribution while remaining focused on the authors' previously published data. But in the absence of such a comparison, several important questions remain unanswered. Is there something special about the Bernoulli clicks task that reveals the bump kernel? If so, what? Is it meaningful that the Brunton DDM can account for it with parameters designed to model nonlinearities in sensory processing (suggesting that the bump kernel is not a feature of the decision-making process per se)? The overall sense of uncertainty is exacerbated by the lack of explanation for the striking individual differences in integration kernels (and by implication, processing time constants). To be fair, the authors sketch a possible explanation, but one that remains rather speculative.

Despite these residual concerns, the revised manuscript is technically strong, centers on important questions in computational neuroscience, and suggests some interesting new ways to think about those questions. I expect that it will motivate future work to resolve some of the outstanding questions.

We appreciate the positive response. In response to the reviewer's remaining concerns about generalizability, specifically about the generalizability to the Poisson Task, we would like to point out that our previous work has shown that the fitted parameter values of DDM in our participants are consistent with those reported by Brunton and colleagues (see Figure below). Therefore, one possible explanation is that behaviour in this kind of task is extremely varied across participants (as suggested by our data), and that the larger number of human participants in our study better sample the whole range of behaviour.

Moreover, we would like to note that the auditory task for human participants in the Brunton study did not use a pure Poisson process to generate the stimulus — the inter-click interval is guaranteed to be at least 30 ms. The stimulus generated from such a process is qualitatively not that different from the stimulus generated in Bernoulli Clicks. This suggests that the bump integration kernel is not, at least not just, a result of the difference between Bernoulli Clicks and Brunton Poisson Clicks.

In addition, Wyart and colleagues have shown that decision weights of incoming pieces of evidence fluctuated with slow cortical oscillations (Wyart et al. 2012). Even though they did not directly observe an uneven integration kernel in their data, their result that decision weights correlate with a slow rhythmic pattern is consistent with the bump kernel observed in our data, suggesting that the bump kernel may generalize to other tasks. Future work investigating the relationship between neural activity and behaviour in our task would further test this idea of synchronization between cortical oscillations and integration kernel. Neural data could also shed light on why we observe such large individual differences in integration kernel (and by implication, processing time) across participants.

There is also the question of how to interpret the result that Brunton et al. model requires not only a decision bound but also sensory adaptation to account for the bump shaped kernel. It would suggest that the individual differences in kernels is caused by a difference at the sensory processing level but not at the decision making level. However, we note that the Brunton model may not be the only variant of DDM that could account for this bump shaped kernel — other potential candidates include DDM with a collapsing bound, DDM with variable drift rate, etc). Future work to answer these question would be to compare these models on different kinds of datasets.

We have put part of the above response into the Discussion section with the following paragraphs.

“Finally, several important questions remain to be answered. The bump shaped kernel is a novel behaviour in our task and stand in contrast to previously published results in humans in a similar auditory clicks task (Brunton et al. 2013), that observes a flat integration kernel. So what is causing this difference? One possible explanation is that behaviour in this kind of task is extremely varied across participants (as suggested by our data), and that the larger number of human participants in our study better sample

the whole range of behaviour. Importantly, our previous work has shown that the fitted parameter values of DDM in our participants are consistent with those reported by Brunton and colleagues (Keung et al. 2019).

In addition, Wyart and colleagues have shown that decision weights of incoming pieces of evidence fluctuated with slow cortical oscillations (Wyart et al. 2012). Even though they did not directly observe an uneven integration kernel in their data, their result that decision weights correlate with a slow rhythmic pattern is consistent with the bump kernel observed in our data, suggesting that the bump kernel may generalize to other tasks. Future work investigating the relationship between neural activity and behaviour in our task would further test this idea of synchronization between cortical oscillations and integration kernel. Neural data could also shed light on why we observe such large individual differences in integration kernel (and by implication, processing time) across participants.

There is also the question of how to interpret the result that Brunton et al. model requires not only a decision bound but also sensory adaptation to account for the bump shaped kernel. It would suggest that the individual differences in kernels is caused by a difference at the sensory processing level but not at the decision making level. However, we note that the Brunton model may not be the only variant of DDM that could account for this bump shaped kernel — other potential candidates include DDM with a collapsing bound, DDM with variable drift rate, etc. Future work to answer these question would be to compare these models on different kinds of datasets.”

Figure: Comparison of fitted DDM parameters in human participants in Keung et al. 2019 (grey) and human participants doing the auditory task in Brunton et al. 2013 (orange). Each dot is a participant.